# Glycaemic control is associated with SARS-CoV-2 breakthrough infections in vaccinated patients with type 2 diabetes

Raffaele Marfella [1,2✉], Celestino Sardu[1], Nunzia D'Onofrio[1], Francesco Prattichizzo [3], Lucia Scisciola[1], Vincenzo Messina[4], Rosalba La Grotta[3], Maria Luisa Balestrieri[1], Paolo Maggi[1], Claudio Napoli[1], Antonio Ceriello[3✉] & Giuseppe Paolisso[1,2]

Patients with type 2 diabetes (T2D) are characterized by blunted immune responses, which are affected by glycaemic control. Whether glycaemic control influences the response to COVID-19 vaccines and the incidence of SARS-CoV-2 breakthrough infections is unknown. Here we show that poor glycaemic control, assessed as mean HbA1c in the post-vaccination period, is associated with lower immune responses and an increased incidence of SARS-CoV-2 breakthrough infections in T2D patients vaccinated with mRNA-BNT162b2. We report data from a prospective observational study enroling healthcare and educator workers with T2D receiving the mRNA-BNT162b2 vaccine in Campania (Italy) and followed for one year (5 visits, follow-up 346 ± 49 days) after one full vaccination cycle. Considering the 494 subjects completing the study, patients with good glycaemic control (HbA1c one-year mean < 7%) show a higher virus-neutralizing antibody capacity and a better CD4 + T/cytokine response, compared with those with poor control (HbA1c one-year mean ≥ 7%). The one-year mean of HbA1c is linearly associated with the incidence of breakthrough infections (Beta = 0.068; 95% confidence interval [CI], 0.032-0.103; $p < 0.001$). The comparison of patients with poor and good glycaemic control through Cox regression also show an increased risk for patients with poor control (adjusted hazard ratio [HR], 0.261; 95% CI, 0.097-0.700; $p = 0.008$). Among other factors, only smoking (HR = 0.290, CI 0.146-0.576 for non-smokers; $p < 0.001$) and sex (HR = 0.105, CI 0.035-0.317 for females; $p < 0.001$) are significantly associated with the incidence of breakthrough infections.

[1] Università degli Studi della Campania "Luigi Vanvitelli", Piazza Luigi Miraglia 2, 80138 Naples, Italy. [2] Mediterranea Cardiocentro, 80122 Naples, Italy. [3] IRCCS MultiMedica, Via Fantoli 16/15, 20138 Milan, Italy. [4] Sant'Anna Hospital, Caserta, Italy. ✉email: raffaele.marfella@unicampania.it; antonio.ceriello@hotmail.com

An unprecedented vaccination campaign has halted the spreading and mortality of SARS-CoV-2. However, despite this success, a sizeable number of vaccinated subjects still develop COVID-19, with some patients also experiencing hospitalization and other serious outcomes[1,2]. Indeed, emerging data showed that subjects fully vaccinated with BNT162b2 can develop SARS-CoV-2 infection[1–3].

Breakthrough cases are characterized by lower titres of neutralizing antibodies during the peri-infection period[1], suggesting that a proper immune response is mandatory for the vaccine's efficacy. Beyond the effect of time on waning antibody titres and the emergency of viral variants escaping pre-existing immunity, also patient-intrinsic characteristics might influence immune responses to COVID-19 vaccines and thus the efficacy of vaccination[4,5]. In particular, male sex, obesity, cardiovascular disease, and type 2 diabetes (T2D) are frequent conditions observed in fully vaccinated patients developing COVID-19[6–10].

Patients with T2D are characterized by an undermined immune response to both natural infections and vaccination[11]. Among other factors, glycaemic control is a key determinant of immune responses in patients with T2D[12], possibly affecting COVID-19 prognosis[13]. Previous preliminary data showed that COVID-19 vaccines induce a "weaker" immunity in T2D patients with poor glycaemic control (PC) compared with both non-diabetic subjects but also compared with T2D patients with good glycaemic control (GC)[14]. However, whether glycaemic control after vaccination is associated with COVID-19 breakthrough infections in fully vaccinated patients with T2D is currently unknown. Thus, we conducted an observational study to explore the possible association between glycaemic control and immune responses to the mRNA-BNT162b2 vaccine and between glycaemic control and the incidence of breakthrough infection in vaccinated patients with T2D. Here we show that glycaemic control is associated with both a lower immune response to vaccination and with an increased incidence of breakthrough infections in patients with T2D.

## Results

**Participants**. 735 patients responded to recruitment, and 607 completed the baseline survey and fulfilled eligibility criteria. The 494 participants completing the follow-up were stratified in 2 groups, using laboratory data obtained at the 5 visits: T2D patients with GC ($n = 196$, HbA1c one-year mean < 7%) and T2D patients with PC ($n = 298$, HbA1c one-year mean ≥ 7%), as evidenced in the STROBE diagram (Fig. 1). The two groups were well-balanced (Table 1), except for BMI, diabetes duration, HDL-cholesterol, and the prevalence of cardiovascular risk factors, which were thus all considered to adjust the successive analysis. Patients were followed up for $346 \pm 49$ days (SD) from the second dose.

**Level of immune-related parameters according to glycaemic control**. All participants were evaluated to assess their neutralization antibody responses and T-cell responses. T2D patients with HbA1c ≥ 7% showed a significantly reduced virus-neutralizing antibody capacity compared with T2D patients with GC, as assessed by the one-year mean of the percentage of neutralization (Fig. 2A). Moreover, in response to stimulation of peripheral blood cells with S-specific peptide pools, we observed a higher CD4+ T/ cytokine response involving type 1 helper T cells in patients with GC compared with PC, as demonstrated by a higher number (one-year mean) of CD4+ T cells expressing interferon (IFN)-γ (Fig. 2B), interleukin (IL)-2 (Fig. 2C), and tumour necrosis factor-(TNF) α (Fig. 2D) (one-year mean for all).

**Relationship between glycaemic control and immune-related parameters**. To substantiate a possible role for glycaemic control in developing proper immune responses to vaccination, we explored the relationship between the one-year mean of HbA1c and the one-year mean of the four tested immune-related parameters through the PRESS statistic. The results showed that the one-year mean of HbA1c has a good degree of correlation with the one-year mean of antibody-mediated neutralization capacity ($R^2$ adapted 0.593, $p < 0.001$) (Fig. 3A) and with the one-year mean of the number of CD4+ T cells expressing IFN-γ ($R^2$ adapted 0.570, $p < 0.001$) (Fig. 3B), IL-2 ($R^2$ adapted 0.503, $p < 0.001$) (Fig. 3C), and TNFα ($R^2$ adapted 0.497, $p < 0.001$) (Fig. 3D), corroborating a possible association between glycaemic control and immune responses to vaccination.

**Variables associated with breakthrough infections**. The monitoring of patients during the follow-up revealed that, when analyzed as a continuous variable through linear regression, the one-year mean of HbA1c was significantly associated with the incidence of breakthrough infections with a $\beta = 0.068$ (95% confidence interval [CI], 0.032–0.103; $p < 0.001$). When glycaemic control was studied as a dichotomous variable, we recorded 31 (10.5%) breakthrough cases among the 298 T2D patients with PC and 7 breakthrough infections among the 196 (3.6%) T2D patients with GC, with a hazard ratio (HR) = 0.261 (CI 0.097–0.700; $p = 0.008$) adjusted for age, sex, BMI, diabetes duration, HDL-cholesterol, cardiovascular risk factors, and active therapies (Fig. 4A, B). Consistently, the levels of the one-year mean of HbA1c in the groups of subjects infected were significantly higher compared with those not infected during the follow-up (Supplementary Fig. 1).

Among other factors, only smoking (HR = 0.290, CI 0.146–0.576 for non-smokers; $p < 0.001$) and sex (HR = 0.105, CI 0.035–0.317 for females; $p < 0.001$) were significantly associated with the incidence of breakthrough infections (Fig. 4A and Supplementary Fig. 2).

**Level of immune-related parameters in the peri-infection period**. To explore whether infected patients were those with the poorer immune responses, we compared the levels of immune parameters in infected vs non-infected patients. Results showed that infected patients had significantly lower levels of both antibody-mediated neutralization capacity and CD4+ T cells expressing IFN-γ, IL-2, and TNFα (Supplementary Fig. 3).

## Discussion

The vaccination campaign against SARS-CoV-2 has abated the incidence of severe outcomes induced by COVID-19[1–3]. However, viral spreading is still high in many countries, likely due to a combination of factors limiting the vaccine's efficacy against breakthrough infections. The intrinsic impossibility of vaccines to produce a 100% coverage against infections, the effect of time on waning antibody titres, and the emergence of viral variants escaping vaccine-induced immunity are likely the major determinants of vaccination failures[4,5]. However, also patient characteristics might influence the incidence of breakthrough infection, as suggested for patients with different comorbidities, including T2D[6–8]. Data presented here suggest that poor glycaemic control during the year after vaccination worsens the immunological response to the BNT162b2 vaccine and might favour SARS-CoV-2 breakthrough infections within T2D patients.

To our knowledge, this is the first report showing the incidence of breakthrough infections specifically in patients with T2D, a condition that per se is held to be a risk factor for the incidence of

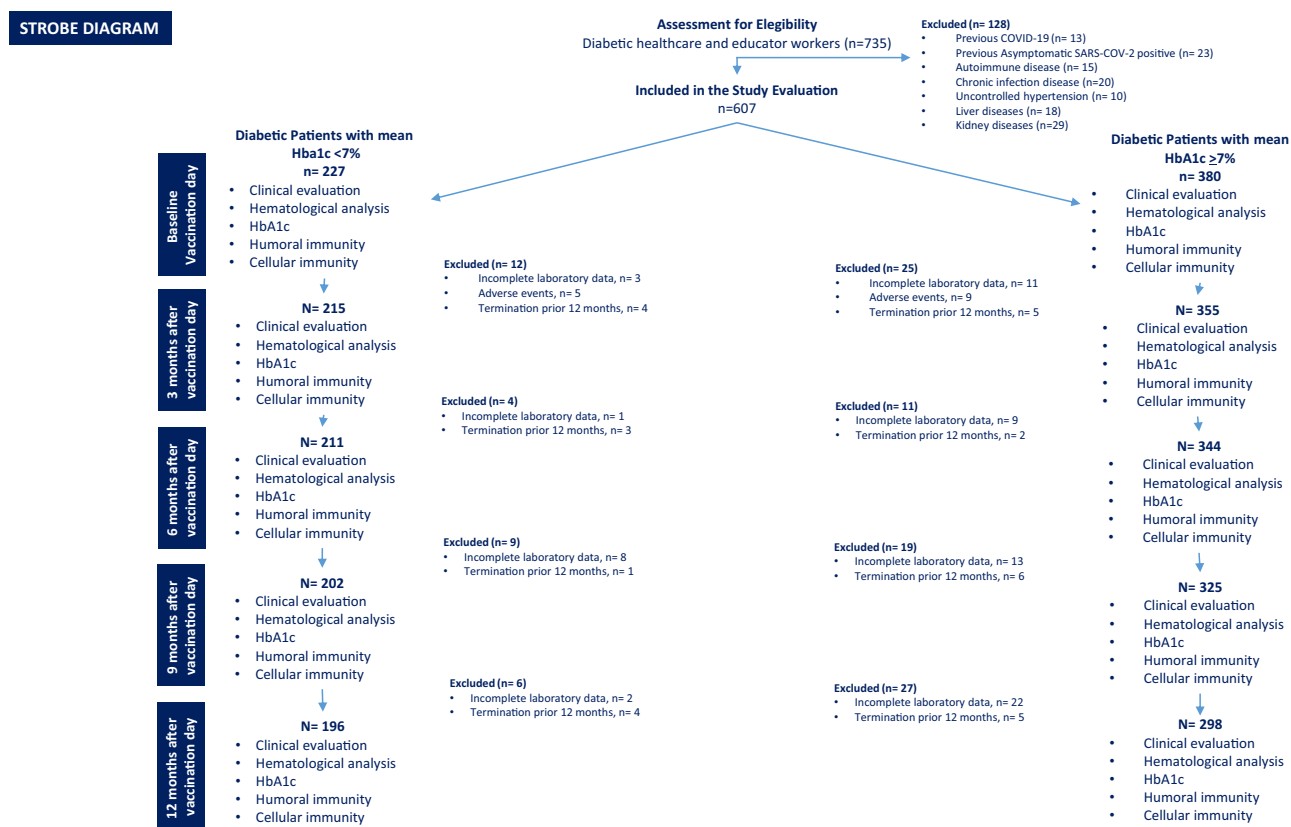

**Fig. 1 Study flow diagram.** STROBE diagram summarizing the flow of the study.

COVID-19[15] and, more broadly, of infectious diseases[16]. Of note, we found that, beyond glycaemic control, the only other factors associated with the incidence of infections were smoking and male sex. Poorly controlled T2D, being male, and smoking have been suggested as major drivers of the dysfunction of the immune system[17–21]. A gradual deterioration of the immune system influences both the host's capacity to respond to infections and the development of long-term immune memory, including proper responses to vaccinations. The adaptive immune system can be affected in terms of impaired proliferation in response to antigenic stimulation, impaired development of CD4+ T follicular helper cells, and a diminished capacity to produce effector lymphokines, among other mechanisms[17–20]. Our findings are compatible with a framework where male T2D patients with poor glycaemic control and smokers are characterized by a more dysfunctional adaptive immune system and are thus less responsive to vaccination and more prone to develop breakthrough infections.

Available data from the general population suggest that patients experiencing infections had weaker immune responses during the peak period[1]. In particular, the antibody neutralization capacity has been reported to be highly predictive of immune protection in vaccinated subjects[22,23]. Our results further support such assumption and extend the observation to cell-mediated immunity.

Our study has limitations. Despite our effort to adjust for a range of risk factors, residual unmeasured confounders are inherently linked to all observational studies. For instance, we did not evaluate eventual behavioural differences among recruited patients. Thus, we cannot exclude that patients with better glycaemic control are also more careful in terms of COVID-19 prevention. Furthermore, our sample size was likely not sufficiently powered to detect the possible association of multiple variables with a small effect, impeding any conclusion relative to factors not associated with breakthrough infections in our cohort. In addition, our study terminated at the end of November 2021, thus before the emergence of the Omicron variant in Italy (https://www.iss.it/web/guest/cov19-cosa-fa-iss-varianti/-/asset_publisher/yJS4xO2fauqM/content/comunicato-stampa-n%C2%B065-2021-identificata-dalla-rete-italiana-una-sequenza-riconducibile-alla-variante-omicron?_com_liferay_asset_publisher_web_portlet_AssetPublisherPortlet_INSTANCE_yJS4xO2fauqM_assetEntryId=5892406&_com_liferay_asset_publisher_web_portlet_AssetPublisherPortlet_INSTANCE_yJS4xO2fauqM_redirect=https%3A%2F%2Fwww.iss.it%2Fweb%2Fguest%2Fcov19-cosa-fa-iss-varianti%3Fp_p_id%3Dcom_liferay_asset_publisher_web_portlet_AssetPublisherPortlet_INSTANCE_yJS4xO2fauqM%26p_p_lifecycle%3D0%26p_p_state%3Dnormal%26p_p_mode%3Dview%26_com_liferay_asset_publisher_web_portlet_AssetPublisherPortlet_INSTANCE_yJS4xO2fauqM_assetEntryId%3D5892406%26_com_liferay_asset_publisher_web_portlet_AssetPublisherPortlet_INSTANCE_yJS4xO2fauqM_cur%3D0%26p_r_p_resetCur%3Dfalse). Thus, we cannot establish if the findings relative to the incidence of infections are valid also for those driven by the new viral strain. However, we showed a consistent relationship between glycaemic control and the immune response to vaccination. This observation should maintain its validity independently of the emergence of viral variants and the redesign of COVID-19 vaccines, as long as the same technology is used. Another limitation of the study is that we recorded only the incidence of SARS-CoV-2 infections and not the development of severe forms of COVID-19. Similarly, data relative to Ct values of positive tests were not available, impeding the exploration of the effect of glycaemic control on viral load, which has been reported to be usually low in breakthrough infections[24,25]. However,

**Table 1 Characteristics of study patients.**

|  | Patients with HbA1c mean < 7% | Patients with HbA1c mean ≥ 7% | P |
|---|---|---|---|
| N | 196 | 298 |  |
| Age, years | 56.9 ± 5.9 | 57.6 ± 6.1 | 0.252 |
| Sex, male, n (%) | 107 (54.6) | 166 (55.7) | 0.284 |
| BMI, kg/m$^2$ | 28.4 ± 1.7 | 29.1 ± 2.1 | **0.002** |
| Diabetes duration, years | 13.1 ± 2.5 | 14.5 ± 3.3 | **0.001** |
| Systolic blood pressure, mmHg | 132.6 ± 14.7 | 132.9 ± 13.7 | 0.787 |
| Diastolic blood pressure, mmHg | 78.6 ± 6.8 | 79.1 ± 6.5 | 0.512 |
| Heart rate, b/min | 75.4 ± 9.9 | 75.8 ± 8.7 | 0.365 |
| Laboratory analysis |  |  |  |
| Blood glucose, mg/dl | 178.8 ± 36.7 | 195.5 ± 28.2 | **0.001** |
| Total cholesterol, mg/dl | 203.8 ± 19.4 | 203.1 ± 22.4 | 0.721 |
| LDL cholesterol, mg/dl | 129.7 ± 18.9 | 128.4 ± 21.6 | 0.511 |
| HDL cholesterol, mg/dl | 38.2 ± 3.5 | 37.1 ± 3.4 | **0.001** |
| Triglycerides, mg/dl | 185.9 ± 21.5 | 187.9 ± 23.3 | 0.331 |
| Creatinine, mg/dl | 0.9 ± 0.15 | 0.9 ± 0.2 | 0.667 |
| HbA1C basal, % | 6.7 ± 0.7 | 7.1 ± 0.7 | **0.001** |
| HbA1c 3 months, % | 6.8 ± 0.4 | 7.6 ± 0.7 | **0.001** |
| HbA1c 6 months, % | 6.9 ± 0.8 | 8.3 ± 1.2 | **0.001** |
| HbA1c 9 months, % | 7.1 ± 0.7 | 8.8 ± 1.9 | **0.001** |
| HbA1c 12 months, % | 6.3 ± 0.76 | 8.2 ± 1.8 | **0.001** |
| Risk factors |  |  |  |
| Hypertension, n (%) | 58 (40.3) | 86 (59.7) | 0.469 |
| Dyslipidaemia, n (%) | 56 (28.6) | 78 (26.2) | 0.314 |
| Cigarette smoking, n (%) | 32 (16.3) | 112 (37.6) | **0.001** |
| Heart disease, n (%) | 41 (20.9) | 71 (23.8) | 0.269 |
| Active therapy |  |  |  |
| Aspirin, n (%) | 54 (27.6) | 86 (28.9) | 0.417 |
| β-Blocker, n (%) | 53 (27.1) | 69 (23.2) | 0.159 |
| Calcium-channel blocker, n (%) | 34 (17.3) | 37 (12.4) | 0.082 |
| Statin, n (%) | 43 (21.9) | 64 (21.5) | 0.494 |
| ACE inhibitor, n (%) | 55 (28.1) | 72 (24.2) | 0.193 |
| AT-2 antagonist, n (%) | 26 (13.3) | 38 (12.8) | 0.485 |
| Oral anti-diabetic drugs, N (%) | 181 (92.3) | 271 (91.1) | 0.458 |
| Insulin, n (%) | 20 (10.2) | 29 (9.7) | 0.489 |

Data are presented as mean ± SD or as number (%). Significant differences are highlighted in bold.
*IHD* ischaemic heart disease, *BMI* body mass index, *HbA1c* Haemoglobin A1c, *HDL* high-density lipoprotein, *LDL* low-density lipoprotein, *ACE* angiotensin-converting enzyme, *AT2* Angiotensin 2.

breakthrough infections have been shown to progress to severe illness at non-negligible rates[1–8]. This observation is particularly relevant for patients with T2D since T2D has been consistently reported to be a risk factor for severe COVID-19[26]. Future studies are warranted to establish if glycaemic control is associated with severe COVID-19 and relative outcomes in vaccinated patients with T2D.

In summary, our findings, coupled with the observation that achieving adequate glycaemic control improves the relative immunological response[14], suggest that the implementation of diabetes care might enhance vaccine effectiveness, thus lowering the risk of SARS-CoV-2 breakthrough infections. While a randomized clinical trial is needed to conclude that lowering HbA1c after vaccination reduces COVID-19 incidence among patients with T2D, these results uncover for the first time a role for glycaemic control as a candidate, key mediator of BNT162b2 vaccine efficacy within the T2D population, suggesting that this specific aspect of diabetes management might contribute to the success of the vaccination programme[27]. In addition, they might further suggest to consider patients with poorly controlled diabetes as a high-risk group to be prioritized for booster doses of vaccine.

## Methods

**Study design.** We conducted a prospective observational study recruiting patients with T2D among subjects involved in the Campania vaccination program for healthcare and educator workers receiving the mRNA-BNT162b2 vaccine (Pfizer-Biotech). Diagnosis of diabetes was based on usual clinical practice in Italy, which

follows guidance from the Standards of Medical Care in Diabetes-2020[28]. Information relative to medicines used for diabetes treatment before the vaccination, the date of the beginning and end of therapy, route of administration, and duration of use were collected. The vaccines were administered as an intramuscular injection into the deltoid. Subjects with clinical or laboratory evidence of previous COVID-19, autoimmune diseases, malignant neoplasms, liver diseases, kidney diseases, uncontrolled hypertension, and chronic infectious diseases were excluded from the evaluation (exclusion criteria).

Subjects included in the study underwent five successive clinical evaluations, routine laboratory analyses, and HbA1c assessments starting from the day of the second vaccine dose (Fig. 1). In addition, neutralization antibody responses and T-cell responses were assessed in all participants after 14 days from the second dose and at each visit.

Finally, we assessed who developed COVID-19 more than 14 days after the second vaccine dose by reverse-transcriptase–polymerase-chain-reaction (RT-PCR) assays performed in all patients at each visit. Monitoring of breakthrough infections was also performed by extensive evaluations of symptomatic healthcare workers (including mild symptoms) or testing of subjects who had exposure to known cases at any time after 14 days from the second dose of vaccine. Patients subjected to the booster dose of vaccine before the completion of the follow-up were excluded from the study.

The study was approved by the Ethic Review Committee of Università degli Studi della Campania "Luigi Vanvitelli" (Protocol no 0029855). Patients signed a written informed consent to use samples for the research and publish the results.

**Real-time reverse transcription PCR assay for SARS-CoV-2.** Respiratory specimens were collected by the local CDC and then shipped to designated authoritative laboratories to detect SARS-CoV-2. SARS-CoV-2 in respiratory specimens was detected by real-time reverse transcription (RT-PCR) methods.

**Assessment of neutralization antibody responses.** To determine the immune status of SARS-CoV-2 vaccinated subjects, GenScript SARS-CoV-2 Surrogate Virus

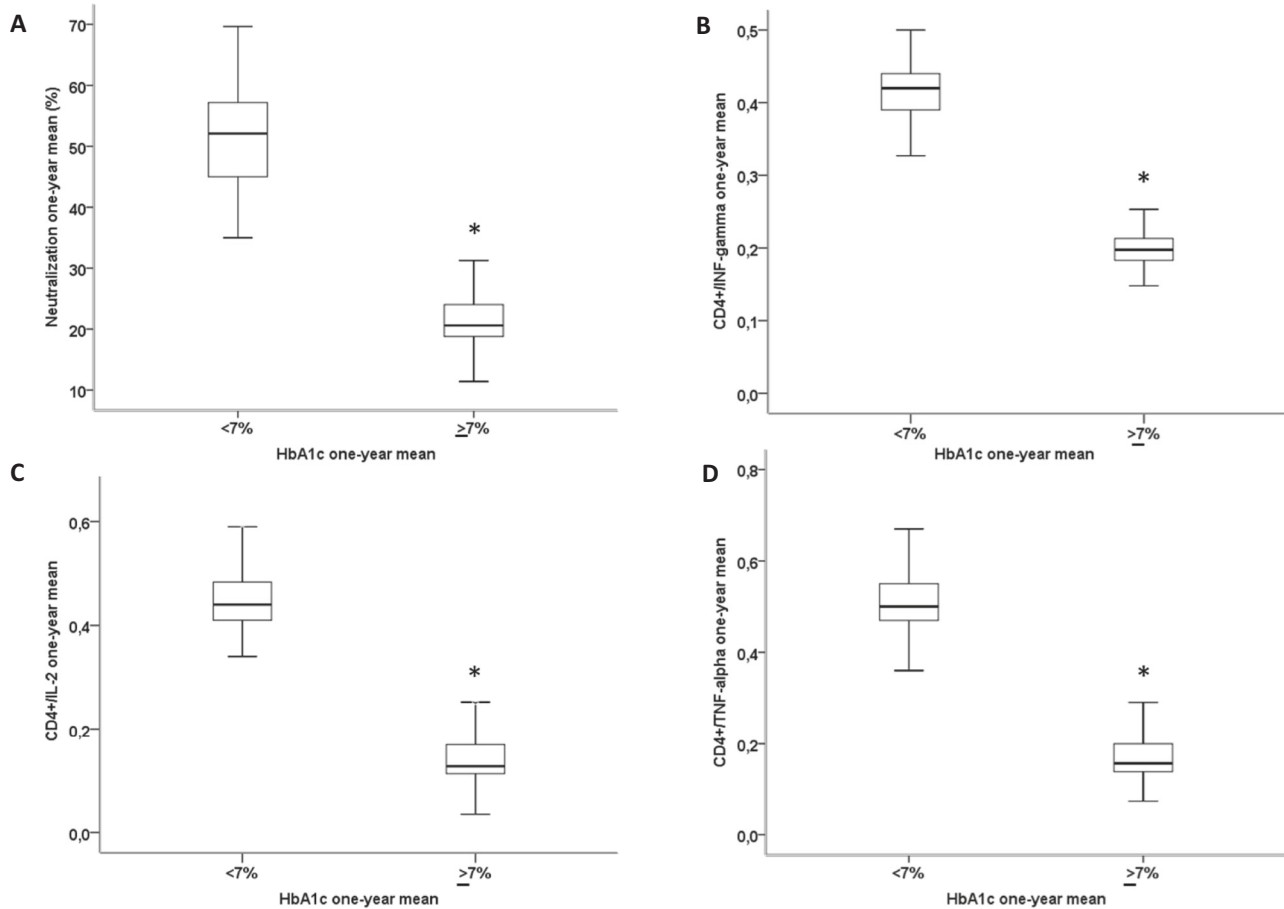

**Fig. 2 Patients with poor glycaemic control have lower immune responses to vaccination.** The one-year mean of neutralizing antibodies capacity (assessed as percentage of neutralization) (**A**), the proportion of CD4+ T cells producing interferon-γ (IFNγ) (**B**), interleukin-2 (IL-2) (**C**), and tumour necrosis factor- α (TNF-α) (**D**), in T2D patients with good glycaemic control (one-year mean HbA1c < 7%, 5 evaluations, n = 196), and in diabetic patients with poor glycaemic control (one-year mean HbA1c ≥ 7%, 5 evaluations, n = 298). Boxplots show the median, 25th, and 75th percentiles, range, and extreme values. *P < 0.01 Two-tailed Student's t test with Bonferroni correction.

Neutralization Test (sVNT) (Cat. No.: L00847-5) was used for neutralizing antibody evaluations. The assay is a blocking ELISA, which mimics this virus receptor binding process to measure the neutralization capacity of anti-SARS-CoV-2 antibodies directed against the receptor-binding domain. Briefly, plasma samples (10 μL) and positive and negative assay controls were incubated with 6 ng of horseradish peroxidase-conjugated RBD (HRP-RBD) at the final dilution of 1:50 at 37 °C for 30 min. The mixtures (100 μl) were then added to the 96-well hACE2-coated capture plate and incubated at 37 °C for 20 min. Next, the supernatant was removed, and the plate was subsequently washed 4x using the provided wash buffer, thus removing the HRP-RBD/neutralizing antibody complexes and allowing unbound HRP-RBD and HRP-RBD/non-neutralizing antibody complexes to remain bound to hACE2. Next, 100 μl tetramethylbenzidine (TMB) was added and incubated for 15 min at room temperature before the reaction was stopped by adding a 50 μl stop solution. The OD of each well was measured by spectrophotometry at 450 nm. Percentage reduction (% reduction) of a sample was calculated as (1-OD450 (sample)/Average OD450 Negative Control) × 100%. Testing each sample in the plasma panel was performed in triplicate, with a quadruplicate deciding test for discordant results.

**Assessment of T-cell responses**

*PBMC isolation.* Briefly, whole blood was collected in a heparin-coated blood bag and, within an hour of their arrival, centrifuged for 10 min at 200 × g to separate the cellular fraction and plasma. Peripheral blood mononuclear cells (PBMC) were isolated using Histopaque-1077 (Sigma Aldrich) by density-gradient sedimentation. Isolated PBMC were cryopreserved in cell recovery media containing 10% DMSO (GIBCO), supplemented with 10% heat-inactivated foetal bovine serum (FBS, 10270-106, Gibco), and stored in liquid nitrogen until used in the assays, as previously described[29]. *Intracellular cytokine staining assay.* Intracellular cytokine-staining assays were performed to quantify antigen-specific T-cell responses against the SARS-CoV-2 spike protein at each planned visit. Briefly, frozen PBMC were thawed, counted, and resuspended in specific culture media (90% RPMI 1640 with

10% FBSand 1% penicillin streptomycin (15140-122, Gibco) and (L-Glutamine 25030-024, Gibco)) overnight at 37 °C with 5% $CO_2$. The day 2 cells were counted, transferred to a 24-well plate(s) ($2.5 \times 10^6$ cells/cm$^2$) and cultured in the presence of PepTivator SARS-CoV-2 Prot_S Complete (130-127-951, Miltenyi Biotec), (1 μg/ml) for 6 h at 37 °C, as reported in manufacturer's protocol. The PepTivator SARS-CoV-2 Prot_S Complete covers the whole protein-coding sequence of the surface or spike glycoprotein S without the first four amino acids of the signal peptide. Following stimulation, cells were washed and stained with viability dye 7-Amino Actinomycin D (7-AAD) (A1310, Thermo Fischer) for 30 min at room temperature. After extracellular surface stain cocktail containing the following antibodies: CD4-FITC (555346, BD biosciences), CD3-PerCP-Cy5-55A (552851, BD biosciences), APC-CD8 (555369, BD biosciences), and V450-CD69 (560740, BD biosciences), cells were fixed and permeabilized using the Cytofix/Cytoperm fixation/permeabilization solution kit (554714, BD biosciences) according to the manufacturer's instructions. Cells were then washed in perm/wash solution followed by intracellular staining (30 min at 4 °C) using the following antibodies: TNF-α-PE (BD biosciences, 554513), IL-2 -PE (BD biosciences, 554566), and IFN-γ-PE (BD biosciences, 554701). Finally, cells were measured on the Accury C6 cytometer (BD biosciences). To identify TNF-α, IL-2, or IFN-γ production on the total population, cells were determined by gating on singlets, lymphocytes, CD3+/viability dye followed by CD4+ or CD8+. Each analyzed cytokine was plotted vs. CD69, and finally, the double CD69+/cytokine+ events were used as positive responses. Gating for positive cytokines was performed using unstimulated samples. An example of gating strategy is reported in Supplementary Fig. 4. All CD4+ T/CD8+ T cells expressing cytokines were reported after background subtraction of identical gates from each individual's negative control stimulation (DMSO). Lymphocytes were analyzed with a FACSAria III (BD Biosciences, San Jose, CA), and data analyzed by FlowJo V10 software (FlowJo LLC, USA).

**Statistical analysis.** Data are presented as mean ± SD. The one-year mean of HbA1c (from 5 successive visits) was calculated and used as the metric of glycaemic

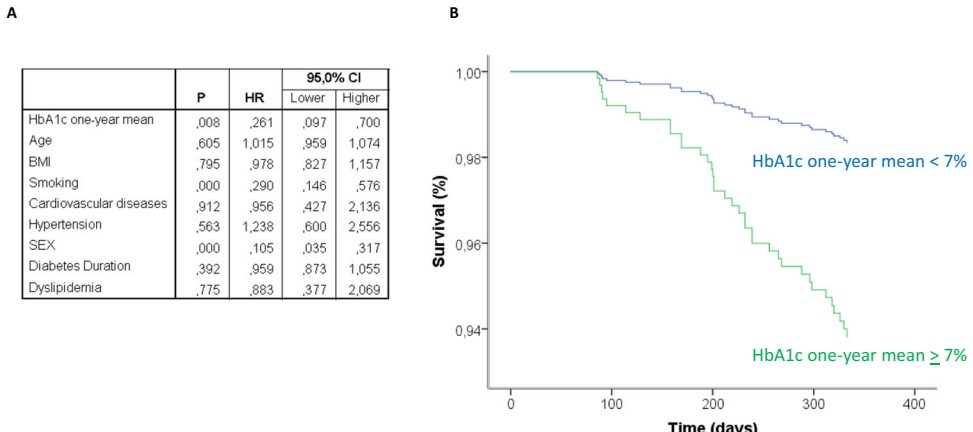

**Fig. 3 HbA1c has a significant relationship with vaccine-related immune parameters.** Regression analysis between neutralization antibodies (**A**), CD4+/interferon-gamma (INF-gamma) (**B**), CD4+/interleukin-2 (IL-2) (**C**), CD4+/tumour necrosis factor-alpha (TNF-alpha) one-year mean (**D**) and HbA1c one-year mean (5 evaluations) predicted residual error sum of squares (PRESS).

**A**

| | P | HR | 95,0% CI | |
|---|---|---|---|---|
| | | | Lower | Higher |
| HbA1c one-year mean | ,008 | ,261 | ,097 | ,700 |
| Age | ,605 | 1,015 | ,959 | 1,074 |
| BMI | ,795 | ,978 | ,827 | 1,157 |
| Smoking | ,000 | ,290 | ,146 | ,576 |
| Cardiovascular diseases | ,912 | ,956 | ,427 | 2,136 |
| Hypertension | ,563 | 1,238 | ,600 | 2,556 |
| SEX | ,000 | ,105 | ,035 | ,317 |
| Diabetes Duration | ,392 | ,959 | ,873 | 1,055 |
| Dyslipidemia | ,775 | ,883 | ,377 | 2,069 |

**B**

**Fig. 4 Poor glycaemic control is associated with an increased incidence of SARS-CoV-2 breakthrough infections.** Results of the Cox regression analysis relative to the association between multiple risk factors, including HbA1c one-year mean, and survival from Covid-19 breakthrough infection, adjusted for age, sex, BMI, diabetes duration, HDL-cholesterol, cardiovascular risk factors, and active therapies (**A**), along with the plot relative to the standardized cumulative incidence of infections in the groups of patients with good glycaemic control (one-year mean HbA1c < 7%, 5 evaluations, $n = 196$) or poor glycaemic control (one-year mean HbA1c ≥ 7%, 5 evaluations, $n = 298$) (**B**).

control to compose (post-hoc) the two groups of poor (PC, HbA1c < 7%) and good glycaemic control (GC, HbA1c ≥ 7%), as suggested by current guidelines[28]. The one-year mean of antibody neutralization and of the number of CD4+ T cells expressing tumour necrosis factor-(TNF) α, interleukin (IL)-2, and interferon (IFN)-γ were used as immune parameters. Continuous variables were compared with Student's t-test for normally distributed data and with Mann-Whitney for non–normally distributed data. When differences were found among the groups,

Bonferroni correction was used to make pairwise comparisons. The Predicted REsidual Sum of Squares (PRESS) statistic was used to explore the relationship between the one-year mean of HbA1c and the one-year mean of the four tested immune-related parameters. Cox regression analysis was used to examine the association between the one-year mean of HbA1c or other risk factors and survival from COVID-19 breakthrough infection and was adjusted for age, sex, BMI, diabetes duration, HDL-cholesterol, cardiovascular risk factors, and active therapies.

The association of the one-year mean of HbA1c with infection was also explored through linear regression analysis using HbA1c as a continuous variable. Only $p$ values of 0.05 or lower were considered statistically significant. All calculations were performed using SPSS 23 software (SPSS Inc, Chicago, IL).

**Reporting summary**. Further information on research design is available in the Nature Research Reporting Summary linked to this article.

## Data availability

The dataset used for the study has been deposited in Zenodo at https://doi.org/10.5281/zenodo.6412110.

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

## Acknowledgements

Programmi di Ricerca Scientifica di Rilevante Interesse Nazionale (Scientific research programmes of high national interest) 2017. N = 2017FM74HK_002. Open Access Funding provided by Universita degli Studi della Campania Luigi Vanvitelli within the CRUI-CARE Agreement. WOA Institution: Universita degli Studi della Campania Luigi Vanvitelli. Blended DEAL: CARE. This work has been also supported by the Italian Ministry of Health - Ricerca Corrente to IRCCS MultiMedica.

## Author contributions

R.M., C.S., F.P., R.L.G., A.C., and G.P. conceived the idea, analyzed data, prepared the figures, and wrote the manuscript. N.D., L.S., V.M., M.L.B., P.M., and C.N. collected data, provided additional expertise, and critically revised the manuscript. The final version of the manuscript was approved by all authors. R.M. is the guarantors of this work and, as such, has full access to all the data in the study and take responsibility for the integrity of the data and the accuracy of the data analysis.

## Competing interests

The authors declare no competing interests.
