## [Peer Review File · Nature Communications]

REVIEWER COMMENTS

Reviewer #1 (Remarks to the Author):

The Paper by Marfella et al is of direct and immediate relevance it shows clearly for the first time that patients with diabetes and glycaemic control is associated with and significant increase in breakthrough infections in fully vaccination against Covid-19.

This is based on some previous data of Israel but is nicely correlated in a prospective study with virus neutralising antibody titers and CD4+T/cytokine response. It confirms previous data showing that diabetes and especially purely controlled diabetes represents a form of an immunosuppressed condition. In their conclusions the authors should stress more clearly that diabetes care should be a key part of managing the current pandemic and post pandemic phase. This should also form a basis for the stratification of booster vaccinations and revaccinations in the high risk groups.

Reviewer #2 (Remarks to the Author):

Marfella et al. report on the impact of glycaemic control on the response to COVID-19 vaccines and on the incidence of SARS-CoV-2 breakthrough infections. This study's results are supported by other former studies' findings. However, I find some major issues in this manuscript that should be addressed before publication:

1) The binary division to 2 classes, namely <7% and >7%, might be delusive and hide the effect of HbA1c levels on breakthrough infections rate. I would kindly suggest the authors re-analyzing the results with continuous variable of HbA1c levels. In addition, I would suggest plotting a figure of the distributions of HbA1c levels in the two groups. I could guess that looking at those distributions will show no justification to this partition (<7%, >7%).

2) As the authors show that the two groups are quite similar in most of their characteristics (except for their HbA1c), one could claim that these differences stem from the same reason: patients that are more careful are in better glycaemic control, and also more cautious about covid-19 and hence being more prone to be infected. Behavioral dissimilarities might dramatically affect the results, and without controlling them carefully I am afraid that these results are not reliable.

3) The study population size is limited and therefore conclusions regarding non-significant factors could mislead: finding variable to be non-significantly affecting is not the same as significantly finding it not affecting (or with negligible effect).

minor:

Line 90: “among those infected, COVID-19 can reach a mortality rate of 22%.” - a meaningless sentence that seems to be out of context, and specifically not inferred from reference #3.

Lines 95-97: sex is also known as a major risk factor for covid-19 (see <https://doi.org/10.1038/s41467-020-19741-6>), and should be taken into account in the analysis.

Lines 189-192: Should refer to the fact that breakthrough infections are known to be with lower viral load compared to unvaccinated infections (see <https://doi.org/10.1038/s41591-021-01316-7> and <https://doi.org/10.1101/2021.02.08.21251329>). Can be included in the analysis of GC also the viral load effect?

Combining the above 2nd and 3rd points, I think that concluding anything about vaccine effectiveness (breakthrough infections), which is vulnerable to behavior and other confounders, is difficult and almost impossible from a small population size.

In conclusion, this manuscript may be of interest to the scientific community. Nevertheless, before publication, the above points should be answered.

Manuscript NCOMMS-22-07259

Response to reviewers' comments

We thank the referees and the editor for the helpful comments on the manuscript. We have addressed all the issues according to the reviewers' suggestions. We believe that these adjustments, elegantly suggested by the reviewers, have improved the paper in terms of clarity and accuracy. Changes are highlighted in the revised version of the manuscript. A clean copy is also attached.

We hope that the revised manuscript is suitable for publication in *Nature Communications*.

Point-by-point responses:

Reviewer #1 (Remarks to the Author):

The Paper by Marfella et al is of direct and immediate relevance it shows clearly for the first time that patients with diabetes and glycaemic control is associated with and significant increase in breakthrough infections in fully vaccination against Covid-19.

This is based on some previous data of Israel but is nicely correlated in a prospective study with virus neutralising antibody titers and CD4+T/cytokine response. It confirms previous data showing that diabetes and especially poorly controlled diabetes represents a form of an immunosuppressed condition. In their conclusions the authors should stress more clearly that diabetes care should be a key part of managing the current pandemic and post pandemic phase. This should also form a basis for the stratification of booster vaccinations and revaccinations in the high risk groups.

We really appreciate the positive evaluation provided by this Reviewer. In the revised version of the manuscript, we stressed the observation relative to diabetes care as a key part of the management of the pandemic and post-pandemic phase, advancing also the hypothesis that patients with poorly controlled diabetes might be included in the high-risk groups to be prioritised for booster vaccinations (page 10, lines 213,214, referring to the file with changes highlighted).

Reviewer #2 (Remarks to the Author):

Marfella et al. report on the impact of glycaemic control on the response to COVID-19 vaccines and on the incidence of SARS-CoV-2 breakthrough infections. This study's results are supported by other former studies' findings. However, I find some major issues in this manuscript that should be addressed before publication:

We really thank the Reviewer for his/her thoughtful comments on our paper. These issues have been now addressed in the revised form of the manuscript.

1. ***The binary division to 2 classes, namely <7% and >7%, might be delusive and hide the effect of HbA1c levels on breakthrough infections rate. I would kindly suggest the authors re-analyzing the results with continuous variable of HbA1c levels. In addition, I would suggest plotting a figure of the distributions of HbA1c levels in the two groups. I could guess that looking at those distributions will show no justification to this partition (<7%, >7%).***

We thank the Reviewer for the precious suggestion. We performed a linear regression to explore the association between the one-year mean of HbA1c, studied as a continuous variable, and breakthrough infections. The results showed a significantly increased risk with increasing HbA1c (Beta = 0.068; 95% confidence interval [CI], 0.032-0.103; $p < 0.001$). In other words, each increase in 1% of HbA1c was associated with an increased 6,8% (3,2% - 10,3%) risk of experiencing a breakthrough infection. We modified the abstract, the methods and the results section to add this result (please see page 3, lines 47-49; and page 7, lines 136-139; and page 13, lines 306-308, referring to the file with changes highlighted). In addition, to cope with the suggestion of this reviewer, we also showed the distribution of HbA1c values in the two groups of patients with vs those without infection (now **Supplementary Figure 1**). Results suggest that infected patients had significantly higher levels of the one-year mean of HbA1c (page 7, lines 145-147).

2. ***As the authors show that the two groups are quite similar in most of their characteristics (except for their HbA1c), one could claim that these differences stem from the same reason: patients that are more careful are in better glycaemic control, and also more cautious about covid-19 and hence being more prone to be infected. Behavioral dissimilarities might dramatically affect the results, and without controlling them carefully I am afraid that these results are not reliable.***

We thank the Reviewer for raising a very important point. The study population was composed of healthcare and educator workers, whom theoretically should have the same caution in managing the risk of infection, given the stringent measures of prevention being applied in Italian hospitals and schools. Indeed, during the whole 2021, the use of face

masks was mandatory everywhere except when outdoor (<https://www.governo.it/it/articolo/domande-frequenti-sulle-misure-adottate-dal-governo/15638#zone>). In addition, the use of FFP2 masks was mandatory in hospitals and highly recommended for educators at school. On the other hand, as suggested by this Reviewer, it is hard to control for such behavioural dissimilarities among groups in observational studies. Thus, we added this aspect as a major limitation of the study (Discussion section: page 9, lines 187-189).

- 3. The study population size is limited and therefore conclusions regarding non-significant factors could mislead: finding variable to be non-significantly affecting is not the same as significantly finding it not affecting (or with negligible effect).**

We agree with the Reviewer's observation. We added the limited sample size and the resulting impossibility to draw conclusions relative to non-significant variables as another limitation of the study (Discussion section: page 9, lines 189-192).

minor:

Line 90: "among those infected, COVID-19 can reach a mortality rate of 22%." - a meaningless sentence that seems to be out of context, and specifically not inferred from reference #3.

We are sorry for the oversight. We removed the suggested sentence.

Lines 95-97: sex is also known as a major risk factor for covid-19 (see <https://doi.org/10.1038/s41467-020-19741-6>), and should be taken into account in the analysis.

We thank the reviewer for the precious suggestion. Accordingly, we re-performed the Cox regression analyses to include also sex as an additional covariate. The results show that also sex is significantly associated with breakthrough infections. These novel data and the modified results relative to the other variables have been added to the text, the figures, the table, and commented in the discussion section (please see **Figure 4A, Table 1, Supplementary Figure 2**, page 3, lines 50-54, and page 7, lines 143,149, and page 8, lines 171,172). In addition, we mentioned sex as a risk factor for COVID-19, taking advantage of the suggested reference (page 5, lines 94-96).

Lines 189-192: Should refer to the fact that breakthrough infections are known to be with lower viral load compared to unvaccinated infections (see <https://doi.org/10.1038/s41591-021-01316-7> and <https://doi.org/10.1101/2021.02.08.21251329>). Can be included in the analysis of GC also the viral load effect?

We thank the reviewer for the helpful comment. Accordingly, we mentioned the observation that breakthrough infections are usually accompanied by a lower viral load (see page 9, lines 199-201), using the two suggested references. However, we have no data relative to the Ct of positive patients since nasopharyngeal swabs were processed in designated authoritative laboratories affiliated with the health national system but not in our lab. We mentioned this aspect as an additional limitation of the study (page 9, lines 199-201).

Combining the above 2nd and 3rd points, I think that concluding anything about vaccine effectiveness (breakthrough infections), which is vulnerable to behavior and other confounders, is difficult and almost impossible from a small population size.

In conclusion, this manuscript may be of interest to the scientific community. Nevertheless, before publication, the above points should be answered.

We thank the reviewer for his/her overall evaluation and for the insightful comments. We believe that we addressed all the major issues suggested and that the paper is tangibly improved after the round of revision.

REVIEWERS' COMMENTS

Reviewer #2 (Remarks to the Author):

I am satisfied with the revision and believe the manuscript is now ready for publication.